# Ethno-veterinary practice for the treatment of animal diseases in Neelum Valley, Kashmir Himalaya, Pakistan

**Sardar Muhammad Rafique Khan** [1]*, **Tanveer Akhter**[1], **Mumtaz Hussain**[2]

**1** Department of Botany, University of Azad Jammu & Kashmir, Muzaffarabad, Pakistan, **2** Department of Botany, Hazara University, Mansehra, Pakistan

* ddcccajk@gmail.com

**Data Availability Statement:** All relevant data are within the paper and its Supporting information files.

**Funding:** The author(s) received no specific funding for this work.

## Abstract

Plant species are not only used as fodder or forage but also contribute substantially in the treatment of various health disorders, particularly in livestock. This study is the first quantitative ethnobotanical effort on ethnoveterinary uses of medicinal plants conducted in the Upper Neelum Valley of Azad Jammu & Kashmir, Pakistan. Information pertaining to cure different ailments of animals were collected from 126 informants through semi-structured interviews, group discussion and field walks. In order to identify the plant species used and their preferred habitats, elderly and experienced members of the tribes, locally known 'Budhair' (aged), were interviewed and sometimes accompanied in the field. The data was further analyzed through ethnobotanical indices. In all, 39 plant species, belonging to 31 genera and 21 families were documented which were used by the indigenous communities of Kashmir Himalaya for curing 21 different diseases of 7 different types of livestock. The highest number of ethno-medicinal plants were contributed by the Polygonaceae family, followed by Crassulaceae, Asteraceae and other families. Roots were the most used part of the plant for preparing ethnoveterinary medicines, followed by the aerial parts. The highest frequency of citation (41) and relative frequency of citation (7.32) was recorded for *Saussurea lappa*, followed by *Rumex acetosa* (37/6.61), *Rumex nepalensis* (36/6.43), *Thymus linearis* (28/5.0) and *Angelica cyclocarpa* (28/5.0). The highest use value was recorded for *Saussurea lappa* (0.33), followed by *Rumex acetosa* (0.29), *Rumex nepalensis* (0.29), *Thymus linearis* and *Angelica cyclocarpa* (0.22 each). The current study has made an important contribution towards the preservation of indigenous plants-based knowledge from extinction. The phytochemical and pharmacological investigations of the plants with high use value can be a potential source of novel drugs to treat health problems of animals and humans.

## Introduction

Medicinal plants have been used across the globe since ages due to their efficacy, availability as well as cultural beliefs. The herbal remedies are an essential part of the traditional medicinal

**Competing interests:** The authors have declared that no competing interests exist.

practices in the indigenous Himalayan mountain communities. Plant based ethnoveterinary medicine are widely practiced in the Himalayan region since the livestock rearing is an integral part of the livelihoods [1]. These traditional herbal medicines provide efficient and cheap therapies along with their common accessibility in comparison to the western allopathic drugs [2]. This ethnic knowledge is directly linked with the local biodiversity and runs deep in the fabric of the rural societies through centuries [3]. Documentation of this altruistic folk knowledge holds key importance especially with the ratification of the Nagoya Protocol in order to maintain cultural heritage [4]. The growing scientific evidence suggests that this Ethnic knowledge supplemented with the new scientific insights can offer socially acceptable and eco-friendly approaches vital for the sustainable development of the local communities [5].

The western Himalayan mountains of Kashmir region supports rich biodiversity attributed to its diverse geography and landscape spanning from deep valley floor through terraced lands and dense forests, up to snow-capped alpine peaks [6]. This mosaic of diverse niches, habitat heterogeneity and the microclimatic variation along the altitudinal gradient results into harboring a bewildering floristic diversity in the region [7]. The rural mountain communities of the Kashmir region practice an agro-pastoral semi nomadic lifestyle, mainly depending on livestock rearing and subsistence agriculture for their livelihood [8].

Medicinal plants have been widely used as a primary source of prevention and control of livestock diseases in the local communities for several centuries, as the inhabitants have learned the medicinal usage of plants growing in their close vicinity [9]. It is an interesting topic to assess the monitory values of this plant based ethno-veterinary linked directly with the increasing cost of livestock rearing and maintenance. Furthermore, these ethnoveterinary medicine are very dynamic and multipurpose as they can treat several different types of livestock disorders, along with being readily available in the remote areas and cheapest as compared to the synthetic drugs [10].

This precious indigenous knowledge has usually been disseminated from one generation without any proper documentation and preservation [11]. The ethnoveterinary knowledge in the region is facing a threat of erosion as the locals are changing their preferences due to rapid socioeconomic transformations in the region synchronized with the environmental changes and technological advancements [12]. The researchers have done a lot of work on the ethno-medicinal applications of plants for human health [7, 13–23]. But literature review reveals that very few studies have been carried out on ethnoveterinary applications of the local herbs in the region indicating a significant knowledge gap [24–31].

Although there are very few studies available on the indigenous ethno-veterinary practices in various parts of Pakistan [22, 32–37], the western Himalayan mountain region of Kashmir still remains unexplored in this regard because of its remoteness, harsh climatic conditions and rugged terrain. Current study was designed to document the valuable ethnoveterinary knowledge from this unexplored area to fill the knowledge gap. The specific objectives of the study include to document the important ethnoveterinary applications of local plant species of the Kashmir region used to treat the livestock ailments and disorders by the mountain populations of the area.

## Materials and methods

### Study area

Natural geomorphological features of Pakistan ranges from the snowcapped peaks of Himalaya and other mountain ranges in the north, the sandy beaches and mangrove swamps in south; allowing different landscapes and climates with variety of flora and fauna. This study was conducted in District Neelum of Azad Jammu & Kashmir (AJ&K), Pakistan, which is a hilly area

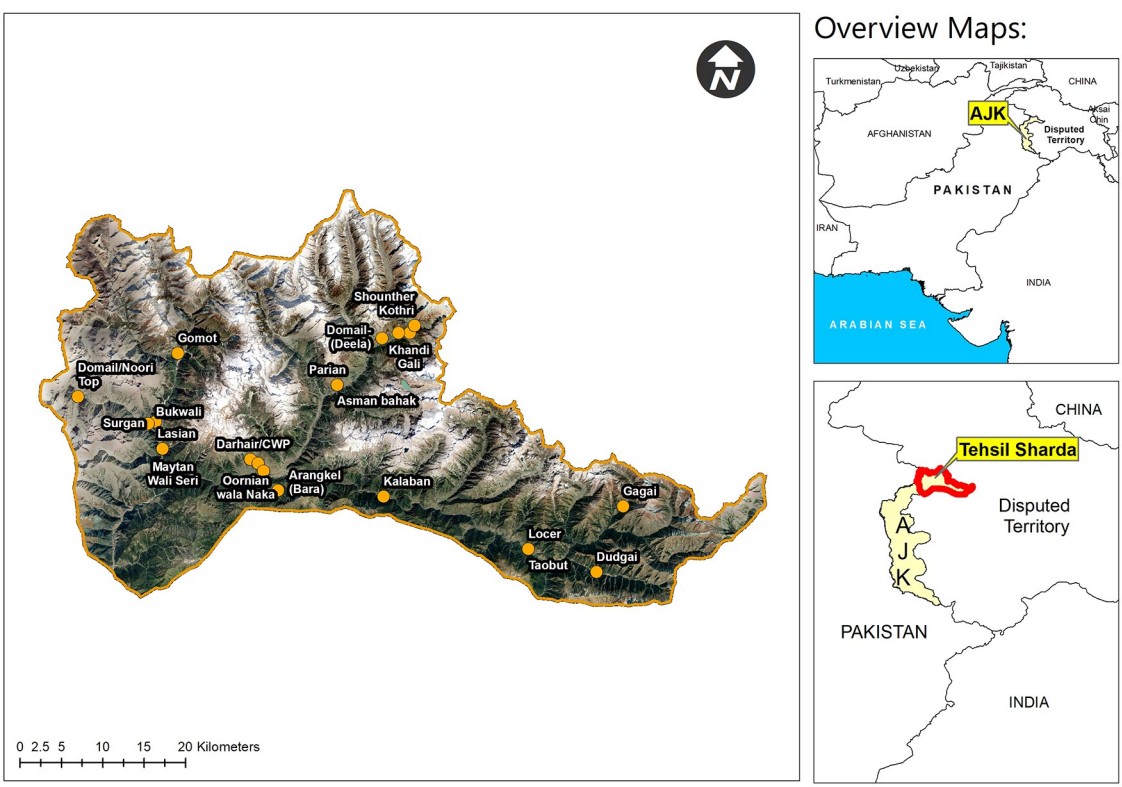

**Fig 1. Map of the study area.**

with rugged topography, located in the extreme north of the AJ&K (Fig 1: Map of the study area). Total area of the district Neelum is 3621 Sq. kms with a population of 1.96 million [37]. Neelum Valley is located at 74˚- 24′–50″ to 74˚–31′–50″ longitude and 34˚–50′–40″ to 35˚ latitudes. Elevation of AJ&K ranges from 360 meters in the south to 6325 meters in the north. The study area lies at an altitude of 2000 meters to 4000 meters. Most of the study area is on high altitude. The climate is temperate with cold winters and moderate summers. The winter season start from November and extends up to April. The high altitude areas remain under snow for 5 months. The major crop of the area is maize, while potatoes and red beans are also cultivated. The valley is rich in the floral diversity. The dominant tree species in the area are *Pinus wallichiana*, *Abies pindrow*, *Picea smithiana*, *Cedrus deodara*, *Acer caesium*, *Aesculus indica* and *Prunus cornuta*, while the dominant shrubs include *Viburnum grandiflorum*, *Indigofera heterantha*, *Rubus ellipticus*. The dominant herbs are *Sambucus wightiana*, *Artemisia vulgaris*, *Lindelofia stylosua*, *Bistorta amplexicaulis*, *Polygonum alpinum* and *Bergenia ciliata*. Neelum Valley is home to different ethnic groups like Mughal, Chaudhry, Butt, Pire, Wani, Syed, Malik, Turks, Khawaja, Rajput, etc. These groups migrated from different areas and are now settled in Neelum Valley. There is cultural and linguistic diversity in the area because of their different past backgrounds. The common languages spoken in the area include Hindko, Kashmiri, Gojri, Shina and Pashtu. The most distinctive features of district Neelum are its mountain ranges, natural lakes, waterfalls and valleys. Documentation was carried out in three sub-valleys of the district Neelum i.e., Surgan, Shounther and Guraize Valley and in a most populated town area Kel (Fig 1: Map of the study area).

There are very limited livelihood opportunities available for the people of Neelum Valley. Most of the pastoralists in the mountain part of Azad Jammu & Kashmir (AJ&K) and the farmers in the high fertile lands are practicing livestock raring from centuries. Livestock plays a pivotal role as it provides farmyard manure, rural transport, milk, meat and source of entertainment in the sports like polo and also has major role in rural economy by providing income and employment to small hold farmers and poor people of the society. Easily accessible and available ethnoveterinary medicinal plants provide a cheaper source for treatment of various diseases. In these communities, the modern veterinary health curative system is inadequate, therefore the inhabitants utilize traditional ethnoveterinary medicinal health system for health care. The economic condition of the farmers also restricts them to the use of modern allopathic drugs, which ultimately leads to poor livestock production and financial losses due to poor health of animals. Under such circumstances, ethnoveterinary medicines can be promoted as an alternative drugs and it can help in alleviation of the poverty by empowering the people to make use of their own resources for the treatment of their livestock.

## Ethnoveterinary field work and interviews

**Ethics statement.**   Code of ethics of International Society of Ethnobiology (2008) was followed during data collection (http://ethnobiology.net/code-of-ethics/). As the data collection was about the animals, therefore, the people who were in close interaction with the animals were targeted. After complete briefings to the informants about the purpose of this research work, verbal consents were taken from all the localities from where the data was collected. As most of the informants were illiterate and it was not possible to take written consent from them.

**Demography and data collection.**   For the collection and documentation of demographic information, well informed persons of the relevant area were approached for interviews and group discussion in accordance with the standardized questionnaires prepared for this purpose. In order to collect the ethnoveterinary information, the data was gathered from the informants, conducting extensive field visits during the year 2012–2015 with the help of pre-planned questionnaires as standardized data collecting protocols [38–40]. Institutional Review Board (IRB) permission was not required for data collection. But formal verbal approval from the respondents was taken before data collection at each locality. The methods employed during the present study were designed with the sole purpose of eliciting the precious wealth of information on the ethno-veterinary uses of medicinal plants practiced by the natives of the Kashmir Himalaya following the methods reported previously [38–40]. Field surveys were conducted in various localities and some of these localities are: Surgan, Kalay Pani, Bukwali, Kel, Arangkel, Domail Bala, Shounther, Lunda Nar, Janawaii, Phulawaii, Halmat and Taobutt, The elderly and experienced members of the tribes, locally known 'Budhair' (aged), preferably above the age of forty were interviewed. More often, they were accompanied to the field for identification of plant species used in the veterinary treatment and their preferred habitats. The survey targeted farmers, shepherds, pastoralists, traditional healers, gardeners, shopkeepers, and plant collectors who had the knowledge of veterinary practices. The plant specimens were shown to them for authentication of relevant information, such as mode of preparation, method of use and dosage of each medicinal plant species. To bring an element of accuracy, the information obtained from one locality was cross-checked with that of others. Distribution status of the plant species used in the veterinary practices in the region (critically endangered, endangered, vulnerable and secure) was also determined on the basis of field observation and information collected from the inhabitants of the area.

**Plant collection, identification and herbarium deposition.** Plant specimens collection and utilization data collection was carried out from upper part of Neelum Valley, located at 74˚- 24′–50″ to 74˚–31′–50″ longitude and 34˚–50′–40″ to 35˚ latitudes and altitude of 6500–13000 feet (2000–4000 meters). Specimens were collected mostly from wild with exception of few (5 cultivated species) from the cultivated lands. There is no requirement of any permit or permission to collect the samples. Most of the collection was carried out from public land which is property of the State and no formal permission is required for research work from the forest department of the State. In case where data collection was required from private lands, verbal permission was sorted from the land owners, before data collection at each site. Specimens of medicinal plants collected from each locality were provided with a collection number for future reference and supported by check lists for inventory. The plant specimens collected were processed at the Herbarium Department of Botany, University of Azad Jammu & Kashmir, Muzaffarabad and then identified with the help of available literature [36, 37, 41–43]. The properly processed plant specimens were deposited in the Herbarium Department of Botany, University of Azad Jammu & Kashmir, Muzaffarabad [43].

**Data analysis.** *Relative frequency citation (RFC).* The frequency of citation was calculated to assess the incidence of one particular plant species used for the treatment of veterinary diseases in relation to the overall citations for all plants. Relative frequency of citation was calculated using RFC = FC/N

Where FC = is the number of informants reporting the use of plant divided by the sum of informants who took part in the study (N) [9].

While, RFC = number of citation (for a given species) divided by number of citations for all species [44].

Frequency of citation for a particular species = (Number of citations for that particular species/Number of all citations for all species)*100.

*Use Value (UV).* Use Value (UV) of a species was calculated using UV = FC/N. where FC is Frequency Citation of one species divided by sum of the informants participated in the study (N).

The relative importance of each species was computed according to the given formula:

$$UVs = \sum \frac{UVi}{Ni},$$

[45]; Where 'UVi" represents use value for a given species among the informants who participated, and 'Ni' represents the sum of informants.

## Results

In the present study, 39 plant species of 21 families have been recorded for their ethnoveterinary importance in the area. A total of 126 informants were interviewed at their homes, in the field or at the religious places through convenience sampling. Among these, 73 were the females and 53 were the male, Young informants (43) were between the ages of 30–45 years, 56 were of the age 40–60 years and sixteen were 61 to 75 years old. Rest of the 11 informants were of the age of 76 or above (Table 1). Majority of the informants (87) were illiterate and 26 informants were having 10 to 12 years of education while 13 informants were holding graduation level degrees. During interviews, it was observed that the illiterate and old age group informants have more traditional knowledge of plants than young and educated class. Females of above 40 years of age were found more informative and true practitioner of the ethnoveterinary sector. All the informants were interviewed in local language Pahari/Hindko/Kashmiri. The key questions on ethnoveterinary were on local names of plants and their parts used,

**Table 1. Informant's demographics in the study area.**

| Gender | Education Level | Occupation | No. of informants |
|---|---|---|---|
| **Female** | Illiterate | Healer | 16 |
| | | Plant collector | 27 |
| | | Shepherd | 14 |
| | Illiterate total | | 57 |
| | Matric / Intermediate | Plant collector | 7 |
| | | Shepherd | 4 |
| | Matric total | | 11 |
| | Graduation | Plant collector | 3 |
| | | Shepherd | 2 |
| | Graduation total | | 5 |
| | **Female total** | | **73** |
| **Male** | Illiterate | Elder Non-professional | 7 |
| | | Farmer | 8 |
| | | Gardener | 2 |
| | | Healers | 3 |
| | | Plant collector | 6 |
| | | Shepherd | 2 |
| | | Healer | 2 |
| | Illiterate total | | 30 |
| | Matric/Intermediate | Elder Non-professional | 3 |
| | | Farmer | 2 |
| | | Gardener | 1 |
| | | Healer | 1 |
| | | Plant collector | 4 |
| | | Shepherd | 1 |
| | | Shopkeeper | 2 |
| | | Trader | 1 |
| | Matric/Intermediate total | | 15 |
| | Intermediate total | | |
| | Graduation | Elder Non-professional | 1 |
| | | Farmer | 1 |
| | | Gardener | 1 |
| | | Healer | 1 |
| | | Plant collector | 2 |
| | | Shepherd | 1 |
| | | Shopkeeper | 1 |
| | Graduation Total | | 8 |
| | **Male total** | | **53** |
| | **Grand total** | | **126** |

mode of preparation and administration, amount of dose given, disease treated and personal experience of informants.

## Taxonomic distribution and growth form of medicinal plants

The current study reported 39 medicinal plants belonged to 21 families, which were used for the treatment of 21 livestock diseases (Table 2). These include 24 herbs (62%), 10 shrubs

**Table 2. Ethnoveterinary use of the plants of Sharda Division, Neelum Valley AJK.**

| S. No. | Plant name | Family | Local name | Habit | Alt. Range (m) | Current Status | Part Used | Ethno-veterinary uses |
|---|---|---|---|---|---|---|---|---|
| 1 | *Aconogonon molle* (D. Don) H. Hara | Polygonaceae | *Chukro* | Herb | 2000–3000 | *Secure* | Rt | Mashed uncooked roots are given orally to cure enterotoxaemia problems (*Andran Da Taap*). Roots after boiling in the water, along with molasses, in solution form are fed orally to cure lamb dysentery (PPR). |
| 2 | *Aconogonon rumicifolium* (Royle ex Rab.) H.Hara | Polygonaceae | *Panchoola* | Herb | 2500–3500 | *Secure* | Rt | Mashed uncooked roots are given orally to cure enterotoxaemia problems. |
| 3 | *Aesculus indica* (Wall. ex Camb.) Hook.f. | Hippocastanaceae | *Bunkhoor* | Tree | 2000–2800 | *Secure* | Frt | Fruits are mashed and fed to the cattle to treat indigestion (*Malla*) as it has warm effect. Seeds are also given orally as tonic, especially to the horses. |
| 4 | *Ajuga bracteosa* Wall. ex Benth. | Lamiaceae | *Jan e Adam* | Herb | 2500–3500 | *Vulnerable* | Rt | Uncooked roots are given orally to the cattle suffering from internal heat (*Peelia*). |
| 5 | *Angelica archangelica* var. *himalica* (Clarke) E. Nasir | Apiaceae | *Murchar* | Shrub | 2000–3000 | *Secure* | Rt | Roots of the plants are cooked and with the addition of molasses are given to cure indigestion cause by the cold. It increases internal temperature and relieve the pain. It is also used to cure dyspnea *(Dhansna)*. |
| 6 | *Angelica cyclocarpa* (Norman) Cannon | Apiaceae | *Chora* | Shrub | 2000–3500 | *Vulnerable* | Rt | Indigestion (locally known as *Dood da Mala*) in cattle is cured by giving uncooked roots with the addition of molasses. Same roots, while cooked are given to the cattle to cure indigestion caused by the cold (locally known as *Thanady da malla*). Used to cure animal's dehydration (*Taku*) issue which usually results because of the non-availability of the water for a long time. |
| 7 | *Aralia cachemirica* Dcne. | Araliaceae | *Chooryal* | Shrub | 1800–2500 | *Secure* | Rt | Mashed and uncooked roots are given to the cattle as tonic which also increases the production of milk. |
| 8 | *Berberis lycium* Royle | Berberidaceae | *Sunmbal* | Shrub | 1800–2700 | *Secure* | Brk | The bark of the root and stem is peeled off, dried, grinded and then used in combination with rice, maize floor and butter as tonic to strengthen the bones and treatment of internal fractures. |
| 9 | *Bistorta amplexicaulis* var. *amplexicaulis* (D. Don) Green | Polygonaceae | *Chiti Masloon* | Herb | 2000–3500 | *Secure* | Rt | Cooked roots (decoction) are given to the feeble cattle as tonic. |
| 10 | *Bistorta amplexicaulis* var. *speciosa* (Meisn.) Munshi & Javeid | Polygonaceae | *Bari Masloon* | Herb | 1800–2500 | *Secure* | Rt | Cooked roots are given to the feeble cattle as tonic. |
| 11 | *Capsicum annuum* L. | Solanaceae | *Rattian Marchan* | Shrub | 1800–2000 | *Cultivated* | Frt | Cotton cloth kept on hay and burnt. Dried fruits of *Capsicum annuum* (locally called *Rattian marchan*) are grinded and the powder in combination with sugar is also poured on the fire. Smoke and fumes produced are being forcibly inhaled to animal (horse/mule/donkey). Consequently, there is copious discharge from nasal cavities and animals become healthy. The disease is known as strangles (locally called *Kannar*). |
| 12 | *Cedrus deodara* (Roxb. ex D.Don) G.Don | Pinaceae | *Pluddar* | Tree | 1800–2250 | *Vulnerable* | Rsn | Resin extracted from the trunk of *Cedrus deodara* after heating the chopped parts of the woods, is applied on the affected skin to cure/mange ecto-parasitism. Burned mobile oil is also used for the same purpose |

(*Continued*)

**Table 2.** (Continued)

| S. No. | Plant name | Family | Local name | Habit | Alt. Range (m) | Current Status | Part Used | Ethno-veterinary uses |
|---|---|---|---|---|---|---|---|---|
| 13 | *Curcuma longa* L. | Zingiberaceae | *Liddhar* | Herb | 1800–2300 | *Cultivated Secure* | Rt | Roots of *Curcuma longa* (Haldi) are cooked in ghee are also fed orally to cure the issue of Prolapse of Uterus. |
| 14 | *Dipsacus inermis* Wall. ex Roxb. | Dipsaceae | *Pilha* | Herb | 1800–2500 | *Secure* | Rt | About 1–2 kg of roots are mashed and cooked in water and given orally to expel placenta as post-delivery treatment in cattle. This has shown quick results and placenta is removed. Cooked roots of *Dipsacus inermis* (Pilha) are also used to cure prolapse of uterus (*Mongra Ana / Bhar Ana*). |
| 15 | *Geranium wallichianum* D.Don. ex Sweet | Geraniaceae | *Ratanjoog* | Herb | 1800–3000 | *Secure* | Rt | Cooked roots are given to the cattle as tonic. |
| 16 | *Helianthus annuus* L. | Asteraceae | *Gul e Aftab* | Shrub | 1800–2300 | *Cultivated Secure* | Sd | Crushed seed (Powder) are given to the week cattle orally as tonic to the general weakness. |
| 17 | *Hylotelephium ewersii* (Ledeb.)H. Ohba | Crassulaceae | *Loonslooni* | Herb | 2500–3500 | *Secure* | AP | Whole mashed uncooked plant is fed to the goats and sheep to reduce the effects of over dozed salts, hence known as *Loonslooni* (*Loon* is local name of Salt). |
| 18 | *Indigofera heterantha var. heterantha* (Wall. ex Baker) Ali | Papilionaceae | *Kainthi* | Shrub | 1800–2800 | *Secure* | Rt | Mashed uncooked roots are given to the young cattle as dewormer. |
| 19 | *Lavatera cachemiriana var. cachemiriana* S. Abdin | Malvaceae | *Dang Sonchal* | Shrub | 2000–2800 | *Vulnerable* | Rt | Cooked roots are used to treat constipation in the animals. |
| 20 | *Ligularia amplexicaulis* DC. | Asteraceae | *Mata Khaish* | Herb | 2800–3600 | *Secure* | Rt | Crushed uncooked roots are given orally to the young cattles to expel worms from the abdomen. It increase digestion and helps the young ones to graze fresh grass. |
| 21 | *Phaseolus lunatus* Linn. | Papilionaceae | *Mooth* | Climber | 1800–2400 | *Cultivated Secure* | Sd | Seeds (Mooth) after boiling in the water are fed, so that blister should appear on the outer surface of the animal to cure Goat Pox (*Thandian*). Otherwise, the death of the animal is possible. |
| 22 | *Phaseolus vulgaris* Linn. | Papilionaceae | *Mooth* | Climber | 1800–2400 | *Cultivated Secure* | Sd | Seeds (Moth) after boiling in the water are fed, so that blister should appear on the outer surface of the animal to cure goat pox (*Thandian*). Otherwise, the death of the animal is possible. |
| 23 | *Punica granatum* L | Lythraceae | *Darru* | Shrub/ tree | - | *Cultivated* | Frt | Other than the *Alum (Phatkri)*, outer fleshy part of the fruit of *Punica granatum* (locally known as *Darru*) is dried, grinded and mixed in yogurt and fed orally to cure the issue of nephritis locally called *Dkahotra/ Chulkna*. |
| 24 | *Rheum webbianum* Royle | Polygonaceae | *Chootyal* | Herb | 2500–3600 | *Vulnerable* | Rt | Mashed roots are given to cure indigestion and constipation issues in cattles. Mashed roots are also tied on the external injuries to relieve pain in the cattles. |
| 25 | *Rhodiola himalensis* (D. Don.)S.H.Fu. | Crassulaceae | *Bugomasti* | Herb | 2700–3600 | *Secure* | AP | Aerial parts are crushed and fed to the young cattle as dewormer. |
| 26 | *Rhodiola pinnatifida* Boiss. | Crassulaceae | *Bugomasti* | Herb | 2800–3600 | *Secure* | AP | Aerial parts are crushed and given to the young cattle as dewormer |
| 27 | *Rhodiola sp.* | Crassulaceae | *Bugomasti* | Herb | 2600–3500 | *Secure* | AP | Aerial parts are crushed and given to the young cattle as dewormer |
| 28 | *Rumex acetosa* L. | Polygonaceae | *Sufaid Hoola* | Herb | 1800–3000 | *Secure* | Rt | Cooked roots are believed effective to cure cough, indigestion and constipation. Roots are buried under the fire in ash and used to expel retained placenta as a post-delivery complication and also on cough. |

(*Continued*)

**Table 2.** (Continued)

| S. No. | Plant name | Family | Local name | Habit | Alt. Range (m) | Current Status | Part Used | Ethno-veterinary uses |
|---|---|---|---|---|---|---|---|---|
| 29 | *Rumex nepalensis* Spreng. | Polygonaceae | *Hoola* | Herb | 1800–3000 | *Secure* | Rt | Cooked roots are believed effective to cure cough, indigestion and constipation. Roots are buried under the fire in ash and used to expel retained placenta as a post-delivery complication and also on cough. It is also used to cure dyspnea (*Dhansna*). |
| 30 | *Saussurea lappa* (Dcne.) Sch. Costus (Falc. Lipsch.) | Asteraceae | *Kuth* | Herb | 2500–3500 | *Critically Endangered* | Rt | Crushed roots are given uncooked to the sheep and goats to expel worms and also believed as tonic. The cattle start eating after the treatment. |
| 31 | *Sedum trullipetalum* H&T. | Crassulaceae | *Loonslooni* | Herb | 2500–3600 | *Secure* | AP | Un cooked, mashed whole plant is given to the goats and sheep to reduce the effects of over dozed salts, hence known as Loon slooni (Loon is local name of Salt). |
| 32 | *Taraxacum laevigatum* (Willd.)DC | Asteraceae | *Hand* | Herb | 1800–3200 | *Secure* | Rt | Mashed uncooked roots are given to cure the post-delivery complication especially to expel retained placenta in the cattle. |
| 33 | *Trigonella foenum-graecum* Linn. | Fabaceae | *Sinji* | Herb | 1800–2600 | *Cultivated Secure* | AP | *Trigonella foenum-graecum* (Maithi) is boiled and fed orally to the animals for the purpose to cure prolapse of uterus. |
| 34 | *Thymus linearis* Benth. | Lamiaceae | *Ajwain/ Bun jamain* | Herb | 2800–3500 | *Endangered* | AP | Decoction of the whole plant with addition of milk, maize flour and molasses is orally fed to the animals suffering from indigestion (*Malla*) and hemoglobinuria (*Rut Mortrna*). |
| 35 | *Urtica dioica* L. | Urticaceae | *Kairi* | Herb | 1800–2800 | *Secure* | Lvs | Vesical palpation with irritation, causing plant *Urtica dioica* (Kari), leaves is practiced which cause irritation in the birth canal and eventually animal is set into heat cycle called Repeat Breeding (*Na Thairna*). |
| 36 | *Verbascum thapsus* L. | Saxifragaceae | *Gadikan* | Herb | 1800–3000 | *Secure* | AP | Leaves are cooked and given to the cattle to relieve pain in case of injury. Broad leaves are also lapped on the injured parts to relieve pain. |
| 37 | *Viburnum cotinifolium* D.Don. | Caprifoliaceae | *Ukloon/ Guch* | Shrub | 1800–2800 | *Secure* | AP | Tips of the plants are collected while starting sprouting and are given uncooked orally to the horses and buffalos to cure constipations. |
| 38 | *Viburnum grandiflorum* Wall. ex DC. | Caprifoliaceae | *Ukloon/ Guch* | Herb | 1800–2800 | *Secure* | AP | Sprouting tips of the plants are collected mashed and given to the horses and buffalos uncooked to cure constipations. |
| 39 | *Zea mays* L. | Poaceae | *Makai* | Herb | 1800–2500 | *Cultivated* | AP | Young plants of *Maize* (Makai) dried under shad are boiled and the hot plant parts are tied on the back of animals (cows & buffalos) which is believed to dry the internal fluid from the body of the animal, which is another type of "indigestion" (locally known as *Linga da Malla*) |

Frt = Fruit, Lvs = Leaves, Rt = Root, AP = Aerial Parts, WP = Whole Plant, Sd = Seeds, Flr = Flowers, Rsn = Resin.

(25%), 3 trees (11%) and 2 climbers (1%). Polygonaceae was the dominant family that contributed 7 species, followed by Crassulaceae (5 species), Asteraceae (4 species), Papilionaceae (3 species) and Lamiaceae, Apiaceae, Caprifoliaceae (2 species each). The remaining 11 families were represented by one species each (Fig 2: Family-wise distribution of the plants used for veterinary treatments).

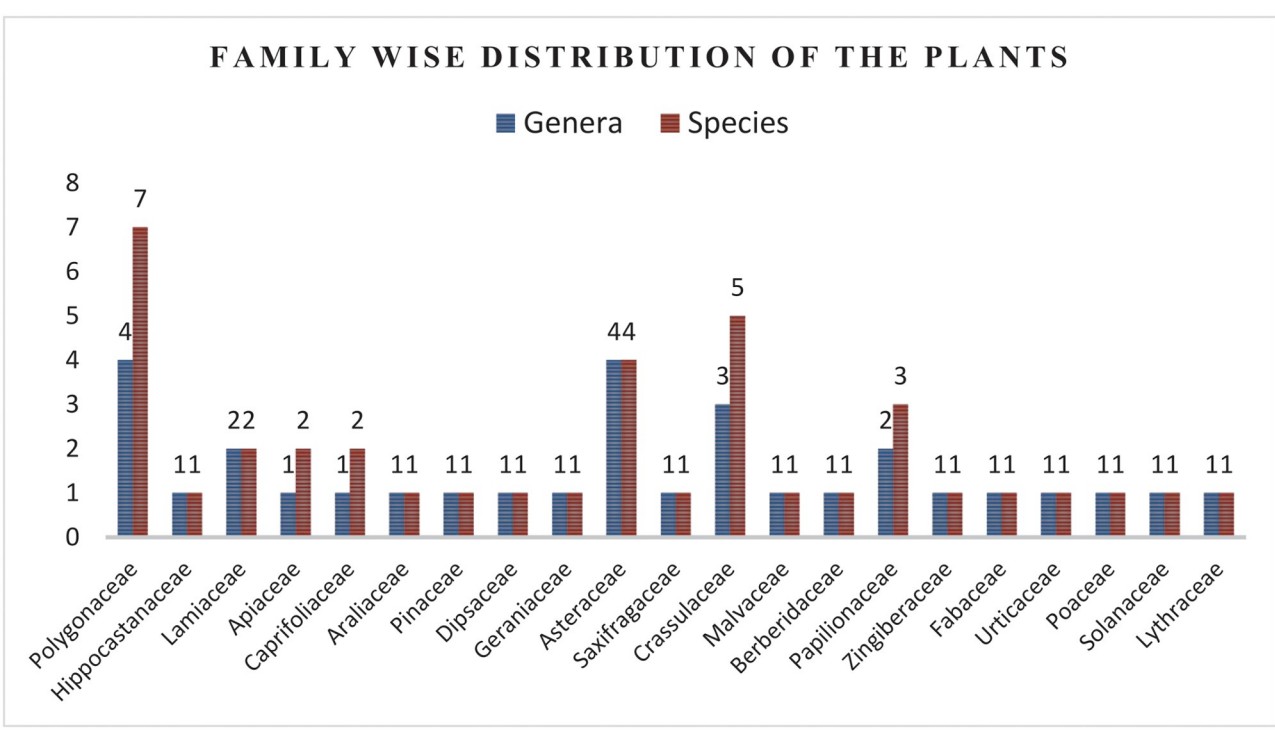

**Fig 2. Family-wise distribution of the plants used for veterinary treatments.**

## Plant part(s) used, formulation and use categories

The information regarding the usage of parts of the plants was obtained from the participants revealed that different parts of the plants are used for preparation of remedies. Roots were the most used parts (49%) followed by aerial parts (28%), seeds (8%), fruits (8%), barks and resins (2% each), and leaves (3%) in the veterinary treatments (Fig 3: Plant parts used to cure different disease in the animals). The main method for preparation of the remedies was mashed uncooked (19 species), cooked (15 species), decoction (03 species), and powder and resin (one species each). The key informants in this study reported 21 major therapeutic uses of the plants which included enterotoxaemia, dysentery, indigestion, internal heat, dehydration, tonic, milk production, ecto-parasitism, post-delivery treatment, anti-salt, hemoglobinuria, prolapse of uterus, Peste des petits ruminants (PPR) a transboundary viral disease, dyspnea, repeat breeding, goat pox, deworming, nephritis, strangles, constipation and cough (Table 2, Fig 4). A total of 9 species were used as tonic, 9 in indigestion, 4 species for post-delivery treatment, deworming and constipations each, 3 for dysentery, 2 for each enterotoxaemia, dyspnea, internal heat, milk production, cough, goat pox, one for anti-salt, dehydration, repeat breeding, nephritis, PPR, strangles, hemoglobinuria and ecto-parasitism.

Medicinal plants used as tonic were *Saussurea lappa*, *Aralia cachemiriana*, *Bistorta amplexicaulis*, *B. affinis*, *Helianthus annus*, *Geranium wallichianum*, *Berberis lycium*, *Aesculus indica* and *Angelica cyclocarpa*. Plant species used for the treatment of indigestion were *Aesculus indica*, *Thymus linearis*, *Saussurea lappa*, *Angelica archangelica*, *A. cyclocarpa*, *Rumex nepalensis*, *Zea mays* and *Viburnum grandiflorum*. Plant species used to cure post-delivery treatments were *Dipsacus inermis*, *Rumex acetosa*, *Rumex nepalensis* and *Taraxacum laevigatum*.

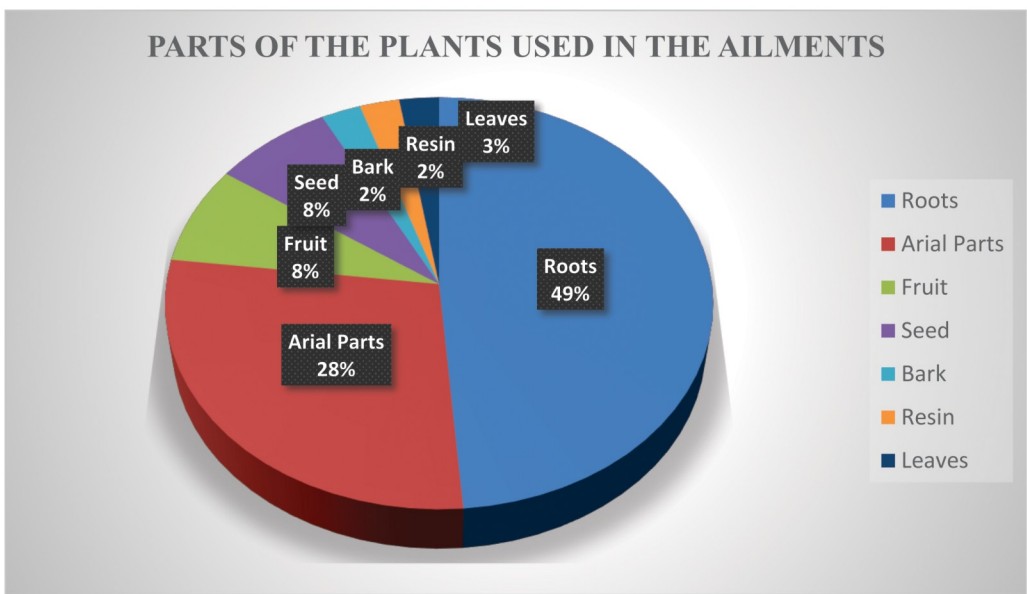

**Fig 3. Plant parts used to cure different disease in the animals.**

Each plant species is provided with its scientific name and author citation, followed by the family, local name (in italics), growth form, altitudinal range in meters above mean sea level), distribution status in the region (critically endangered, endangered, vulnerable and secure), and lastly in brief the part (s) used and the mode of preparation and the dosage (wherever

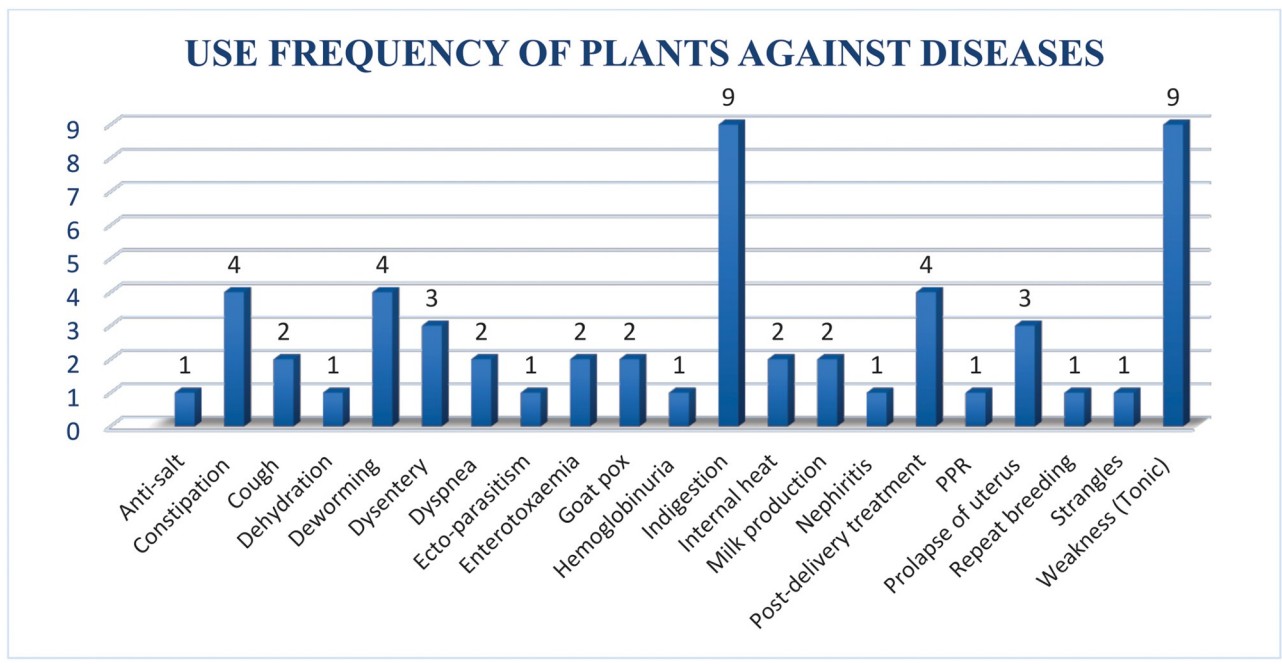

**Fig 4. Frequency of the plant species used against different disease categories.**

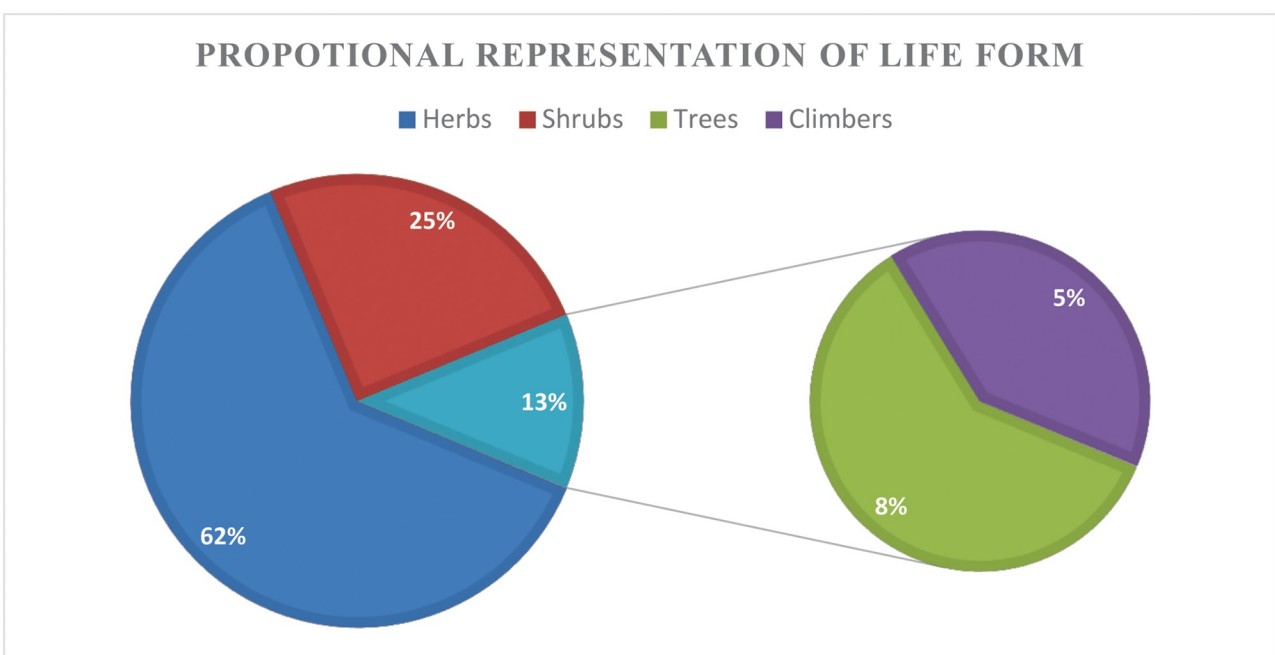

**Fig 5. Proportion of life form of the plant species used in ethnoveterinary.**

available). Proportion of the life form of the species is also given (Table 2; Fig 5: Proportion of life form of the plant species used in ethnoveterinary).

## Relative frequency of citation and use value

Relative Frequency of Citation (RFC) and Use Value (UV) of the medicinal plants was calculated ranging from 41 to 7.32 (Table 3). The highest RFC was found for *Saussurea lappa* (7.32), followed by *Rumex acetosa* (6.61), *Rumex nepalensis* (6.43), *Thymus linearis* (5.0) and *Angelica cyclocarpa* (5.0). The lowest relative frequency of citation was recorded by *Rhodiola pinnatifida*, *Taraxacum laevigatum* and *Helianthus annuus* (0.89 each). The highest UV was recorded for *Saussurea lappa* (0.33), followed by *Rumex acetosa* (0.29), *Rumex nepalensis* (0.29), *Thymus linearis* and *Angelica cyclocarpa* (0.22 each). The lowest use value was recorded in *Rhodiola pinnatifida* and *Taraxacum laevigatum*, which was 0.04 each (Table 3).

## Discussion

Ethnoveterinary applications of the local plant species is an important part of the Himalayan mountain populations in the Kashmir region as livestock rearing plays a vital role in the local microeconomics and livelihood support in the region. Semi nomadic populations prefer the ethno-medicine as compared to the allopathic remedies as they are cheaper and readily available [46]. Our findings revealed that the local populations use a significant number (i.e. 39 spp.) of locally available plants for their livestock health care (Table 2). Medicinal Plant species utilized for livestock treatments harbor diverse range of habitats ranging from valley plains, temperate mountain forests, and alpine pastures climates in a wide altitudinal of 1800–3700 m. [22, 47]. It was observed that old age population groups, especially females, possessed more ethnobotanical knowledge because of their higher association with typical agro-pastoral lifestyle as compared to the younger generation [48].

**Table 3. Relative frequency of citation and use value of plants of the study area.**

| S. No. | Plant name | FC | UV | RFC |
|:---:|:---|:---:|:---:|:---:|
| 1 | *Aconogonon mole* | 25 | 0.20 | 4.46 |
| 2 | *Aconogonon rumicifolium* | 15 | 0.12 | 2.68 |
| 3 | *Aesculus indica* | 19 | 0.15 | 3.39 |
| 4 | *Ajuga bracteosa* | 13 | 0.10 | 2.32 |
| 5 | *Angelica archangelica* var. *himalica* | 20 | 0.16 | 3.57 |
| 6 | *Angelica cyclocarpa* | 28 | 0.22 | 5.00 |
| 7 | *Aralia cachemirica* | 18 | 0.14 | 3.21 |
| 8 | *Berberis lycium* | 16 | 0.13 | 2.86 |
| 9 | *Bistorta amplexicaulis* | 12 | 0.10 | 2.14 |
| 10 | *Bistorta amplexicaulis* var. *speciosa* | 13 | 0.10 | 2.32 |
| 11 | *Capsicum annuum* | 9 | 0.07 | 1.61 |
| 12 | *Cedrus deodara* | 9 | 0.07 | 1.61 |
| 13 | *Curcuma longa* | 7 | 0.06 | 1.25 |
| 14 | *Dipsacus inermis* | 27 | 0.21 | 4.82 |
| 15 | *Geranium wallichianum* | 14 | 0.11 | 2.50 |
| 16 | *Helianthus annuus* | 5 | 0.04 | 0.89 |
| 17 | *Hylotelephium ewersii* | 6 | 0.05 | 1.07 |
| 18 | *Indigofera heterantha* var. *heterantha* | 7 | 0.06 | 1.25 |
| 19 | *Lavatera cachemiriana* var. *cachemiriana* | 11 | 0.09 | 1.96 |
| 20 | *Ligularia amplexicaulis* | 18 | 0.14 | 3.21 |
| 21 | *Phaseolus lunatus* | 6 | 0.05 | 1.07 |
| 22 | *Phaseolus vulgaris* | 7 | 0.06 | 1.25 |
| 23 | *Punica granatum* | 6 | 0.05 | 1.07 |
| 24 | *Rheum webbianum* | 12 | 0.10 | 2.14 |
| 25 | *Rhodiola himalensis* | 6 | 0.05 | 1.07 |
| 26 | *Rhodiola pinnatifida* | 5 | 0.04 | 0.89 |
| 27 | *Rhodiola sp.* | 6 | 0.05 | 1.07 |
| 28 | *Rumex acetosa* | 37 | 0.29 | 6.61 |
| 29 | *Rumex nepalensis* | 36 | 0.29 | 6.43 |
| 30 | *Saussurea lappa* | 41 | 0.33 | 7.32 |
| 31 | *Sedum trullipetalum* | 6 | 0.05 | 1.07 |
| 32 | *Taraxacum laevigatum* | 5 | 0.04 | 0.89 |
| 33 | *Trigonella foenum-graecum* | 7 | 0.06 | 1.25 |
| 34 | *Thymus linearis* | 28 | 0.22 | 5.00 |
| 35 | *Urtica dioica* | 17 | 0.13 | 3.04 |
| 36 | *Verbascum thapsus* | 7 | 0.06 | 1.25 |
| 37 | *Viburnum cotinifolium* | 14 | 0.11 | 2.50 |
| 38 | *Viburnum grandiflorum* | 15 | 0.12 | 2.68 |
| 39 | *Zea mays* | 7 | 0.06 | 1.25 |

The taxonomic analysis indicated the dominance of Polygonaceae, Asteraceae and Crassulaceae (Table 2). These families comprised mostly of herbaceous taxa in the local ethnoveterinary flora which relates with broader ecological amplitude and abundance of these families in the region [49]. The routes of administration of these herbal remedies were essentially oral whereas plant root was the most widely used part followed by the aerial part as a whole or the leaves. The herbs were the leading growth form of the medicinal species followed by shrubs,

and trees (Fig 5). Herbs are often used because of their frequent availability, ease of collection and applications [50, 51]. Plant species were reported to be used through different modes of preparation to form crude drugs as well as to be fed as food supplements to promote faster weight gain, as enterotoxaemia, indigestion, dehydration, ecto-parasitism, post-delivery complications, dewormer, relieve constipation, respiratory, and reproductive disorders [52, 53].

The quantitative ethnobotanical indices offer accurate estimates of the plant use frequencies which can be utilized for the conservation management of the heavily consumed threatened plants of the region [54]. Our results have identified several important plants including *Saussurea lappa*, *Aconogonon molle*, *Angelica cyclocarpa*, *Rumex acetosa*, *Geranium wallichianum* *Rumex nepalensis*, *Angelica glauca* and *Thymus linearis*, having relatively higher use values in the region. Relative Frequency of Citation (RFC) and Use Value (UV) shows that the highest RFC was found for *Saussurea lappa*, *Rumex acetosa* and *Rumex nepalensis* while the lowest relative frequency of citation was recorded for *Rhodiola pinnatifida*, *Taraxacum laevigatum* and *Helianthus annuus*. Similarly, the highest UV was recorded for *Saussurea lappa*, *Rumex acetosa*, *and Thymus linearis* and lowest use value recorded was in *Rhodiola pinnatifida* and *Taraxacum laevigatum* (Table 3) [36, 55–57]. These overexploited species are most prime candidate for conservation in the region demanding immediate attention [55].

It was observed that the method of administering ethno-veterinary plant remedies varied greatly among the different ethnic communities [56]. Different communities were recorded to use different plant species for treating the same disease and vice versa. Similarly plant were used singly, as well as in combinations for treating various livestock ailments which reflects diversity of the ethnic knowledge and heterogeneity in the cultural practices [57].

Ethnic usage of indigenous medicinal plants to treat veterinary disorders and ailments offers a significant contribution in sustaining the livelihood support system of the local populations in the region [58]. The diverse ethnic knowledge reflects the rich cultural values of the society linked with sustainable utilization of the local plant diversity [59, 60]. Results provide a valuable database which has dynamic implications in the management of natural resource in the area [45, 59–69]. These findings also provide baseline information by identifying the effective herbal remedies for livestock health which can be utilized by veterinarians and pharmacologists for the development of new therapies as well as isolation of bioactive compounds [45, 59–70]. The results also serve as a conservationist's proxy and provide an insightful scientific information for the conservation management of overexploited plant species of the region [45, 60, 70, 71].

## Conclusion

Indigenous communities in Neelum Valley are dependent on medicinal plants for ethnoveterinary use. The people practiced 39 medicinal plants to cure 21 livestock diseases. Knowledge about the traditional medicinal system is restricted to the herders, farmers and elder community member. Some important plants like *Dipsacus inermis*, *Rumex nepalensis*, *Angelica cyclocarpa*, *Saussurea lappa*, *Aesculus indica*, etc. are having great significance in the ethnoveterinary practices. Among these, *Saussurea lappa* and *Rumex nepalensis* were found with highest use value and frequency of citation. The younger generation is unaware of this traditional treasure and takes no interest due to modernization. The current study has an important contribution towards the preservation of indigenous plants-based knowledge from extinction. New ethnoveterinary uses in the study area were found for enterotoxaemia, dehydration, indigestion, dewormer, etc. The phytochemical and pharmacological investigations to isolate the active compound and testing the *in vitro* or *in vivo* efficiency of the above mentioned plants against the targeted veterinary diseases are important. In addition to this, critical

toxicological investigations are required for safe and secure use of documented ethno-medicines.

## Supporting information

**S1 File. Sample of questionnaire used during field survey for obtaining ethnobotanical information.**
(DOCX)

**S1 Fig.**
(JPG)

## Acknowledgments

The authors would like to thank Dr. Munir Ahmed for his technical inputs, the interviewees and other inhabitants of Neelum valley, AJ&K, for their contribution to this work.

## Author Contributions

**Conceptualization:** Sardar Muhammad Rafique Khan, Tanveer Akhter.

**Data curation:** Sardar Muhammad Rafique Khan, Mumtaz Hussain.

**Formal analysis:** Sardar Muhammad Rafique Khan, Mumtaz Hussain.

**Investigation:** Sardar Muhammad Rafique Khan, Tanveer Akhter, Mumtaz Hussain.

**Methodology:** Sardar Muhammad Rafique Khan, Mumtaz Hussain.

**Project administration:** Sardar Muhammad Rafique Khan.

**Resources:** Sardar Muhammad Rafique Khan.

**Supervision:** Tanveer Akhter.

**Validation:** Sardar Muhammad Rafique Khan.

**Writing – original draft:** Sardar Muhammad Rafique Khan.

**Writing – review & editing:** Sardar Muhammad Rafique Khan, Tanveer Akhter.

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
