## [Decision Letter · Decision Letter 0]

7 Dec 2020

PONE-D-20-31879

ETHNO-VETERINARY PRACTICES USED FOR THE TREATMENT OF ANIMAL DISEASES IN NEELUM VALLEY, KASHMIR HIMALAYA

PLOS ONE

Dear Dr. Rafique,

Thank you for submitting your manuscript to PLOS ONE. After careful consideration, we feel that it has merit but does not fully meet PLOS ONE’s publication criteria as it currently stands. Therefore, we invite you to submit a revised version of the manuscript that addresses the points raised during the review process.

See my comments on file (2020-Ethno-veterinary Paper _Editor Comments.docx)

We look forward to receiving your revised manuscript.

Kind regards,

Khawaja Shafique Ahmad, Ph.D.

Academic Editor

PLOS ONE

Journal Requirements:

2. Please ensure that you refer to Figure 1 and 4 in your text as, if accepted, production will need this reference to link the reader to the figure.

3. Please upload a copy of Figure 4, to which you refer in your text on page 15. If the figure is no longer to be included as part of the submission please remove all reference to it within the text.

4.We note that [Figure(s) 5] in your submission contain map/satellite images which may be copyrighted. All PLOS content is published under the Creative Commons Attribution License (CC BY 4.0), which means that the manuscript, images, and Supporting Information files will be freely available online, and any third party is permitted to access, download, copy, distribute, and use these materials in any way, even commercially, with proper attribution. For these reasons, we cannot publish previously copyrighted maps or satellite images created using proprietary data, such as Google software (Google Maps, Street View, and Earth). For more information, see our copyright guidelines: http://journals.plos.org/plosone/s/licenses-and-copyright.

1.    You may seek permission from the original copyright holder of Figure(s) [5] to publish the content specifically under the CC BY 4.0 license. 

5. We note you have included a table to which you do not refer in the text of your manuscript. Please ensure that you refer to Table 3 in your text; if accepted, production will need this reference to link the reader to the Table.

6. Please include a copy of Table-4 and Table 5 which you refer to in your text on page 24 and 22.

7.We noticed you have some minor occurrence of overlapping text with the following previous publication(s), which needs to be addressed:

- https://scialert.net/fulltext/?doi=ajps.2007.148.152

The text that needs to be addressed involves the Introduction, the caption for Figure 1, and the final paragraph of the Discussion.

In your revision ensure you cite all your sources (including your own works), and quote or rephrase any duplicated text outside the methods section. Further consideration is dependent on these concerns being addressed.

Additional Editor Comments (if provided):

Please see my comments in track changes

Reviewers' comments:

Reviewer's Responses to Questions

**Comments to the Author**

1. Is the manuscript technically sound, and do the data support the conclusions?

Reviewer #1: Yes

Reviewer #2: No

Reviewer #3: Yes

Reviewer #4: Yes

2. Has the statistical analysis been performed appropriately and rigorously? 

Reviewer #1: Yes

Reviewer #2: N/A

Reviewer #3: Yes

Reviewer #4: Yes

3. Have the authors made all data underlying the findings in their manuscript fully available?

Reviewer #1: Yes

Reviewer #2: Yes

Reviewer #3: Yes

Reviewer #4: Yes

4. Is the manuscript presented in an intelligible fashion and written in standard English?

Reviewer #1: No

Reviewer #2: No

Reviewer #3: Yes

Reviewer #4: Yes

5. Review Comments to the Author

Reviewer #1: The paper has been revised and following correction are recommended.

1. The abstract is very bulky and subjective. The text in the abstract should be reduced/trimmed.

2. The last paragraph of the introduction section given after the objectives should be shifted to discussion. It is not appropriate to give text after stating your specific objectives. The said paragraph also needs to be supported with appropriate references.

3. A brief detail of climate should be given in the methodology/study area section.

4. The 1st paragraph of the results section should be shifted to the methodology as it is not the results.

5. The tables and figures should be cited in the results text.

6. Authors should correlate their own results to the text given in the discussion section.

7. The conclusion section is also bulky, just like abstract and needs reduction in bulk.

Reviewer #2: Review comments of manuscript PONE-D-20-31879

The manuscript presented the ethnoveterinary knowledge of plants, which is used by the local inhabitants in Neelum Valley, Kashmir Himalaya for various disease of animals. The most important issue is that the manuscript was found to have a very significant degree of overlap with existing articles. I have identified a large sections in the manuscript that seem to be taken verbatim from other sources and without giving proper references to these sources and without identifying the text as a word by word citation.

I dont recommend its publication in the form in which it is presented my recommendation is reject and resubmit, I therefore kindly request the authors to provide a substantially revised version of this manuscript to address these concerns and some others suggestions, which is annotated directly in PDF. Note that not all matches in your report will be of concern, in particular commonly found phrases in manuscripts will be highlighted regardless and can be ignored (e.g., acknowledgments section and technical language in method section).

Some others basics missing which I have found and that in each case are necessary for a sound publication as:

The language is not suitable for publication; the manuscript needs a major English revision before being submitted. Formatting of the paper is very poor. Check the manuscript thoroughly for spelling mistake and grammar.

The abstract is not clear and with many serious grammar mistakes.

Your introduction needs more detail. I suggest that you improve the description in the last lines to provide more justiﬁcation for your study (speciﬁcally, you should expand upon the knowledge gap being ﬁlled). Also, write the clear objectives of the study.

The methodology section is poorly organized. No information is available about how the study was design etc.

The results is very poorly presented.

Discussion is not result based, also need improvement grammatically. The whole section has to be revised not only for the language, but also for its structure.

Formatting of tables are in poor order. Please provide voucher specimens number in the table.

The conclusion is overstated, and the major results of the research were not reported.

Critically check your references, some references have missing volume number and page number. Follow Plos One journals guidelines for all references.

There are more annotations in the Pdf, which should be considered by the authors.

Reviewer #3: 1.please shorten the abstract. Such as it went into too many details of the informers.

2.Pay attention to reference 64 in the introduction part. Aren’t they labeled by number orders?

3.Study area in the materials and methods should be clear. I do not understand the necessarity of the first sentence. The investigation was done in a small part of the Neelum Valley. Do they really represent the whole area? If not, you should point out the actual area you covered. Only about a hundred informers were collected, compared to over 1 million population. That is really a small sampling. It would be nice if the sampling area is labeled in the figure 5.

4.To make the best use of the information generated in this paper, I would suggest to compare the medicine with other papers that generated from nearby area in the country.

Reviewer #4: The manuscript is well written but still needs substantial revision and English correction. Author knowledge regarding the study seems to be very narrow and lacks innovative interpretation. More literature study not restricted to study area only but also the Himalaya region is the most. See attachment.

6. PLOS authors have the option to publish the peer review history of their article (what does this mean?). If published, this will include your full peer review and any attached files.

Reviewer #1: **Yes: **Zia-ur-rehman Mashwani

Reviewer #2: No

Reviewer #3: No

Reviewer #4: No

---

## [Author Response · Author response to Decision Letter 0]

6 Jan 2021

Yes I have view this letter and responding accordingly

---

## [Editor Report · Decision Letter 1]

20 Jan 2021

PONE-D-20-31879R1

ETHNO-VETERINARY PRACTICES FOR THE TREATMENT OF ANIMAL DISEASES IN NEELUM VALLEY, KASHMIR HIMALAYA, PAKISTAN

PLOS ONE

Dear Sardar Rafique Khan,

Thank you for submitting your manuscript to PLOS ONE. After careful consideration, we feel that it has merit but does not fully meet PLOS ONE’s publication criteria as it currently stands. Therefore, we invite you to submit a revised version of the manuscript that addresses the points raised during the review process.

ACADEMIC EDITOR:

I have significant concerns about the grammar, usage, and overall readability of the manuscript.  Furthermore, many shortcoming still exist in your manuscript which need to be addressed properly. You can find my comments on file (Manuscript with editor comments).  I believe that the manuscript is much more likely to be accepted if it is easy to read and understand. We therefore request that you revise the text to fix the grammatical errors and improve the overall readability of the text before we can reach any decision.

Please submit your revised manuscript by 19 February If you will need more time than this to complete your revisions, please reply to this message or contact the journal office at plosone@plos.org. Please include the following items when submitting your revised manuscript:

We look forward to receiving your revised manuscript.

Kind regards,

Khawaja Shafique Ahmad, Ph.D.

Academic Editor

PLOS ONE

Additional Editor Comments (if provided):

After assessing your manuscript, I have found significant concerns about the grammar, usage, and overall readability of the manuscript. I believe that the manuscript is much more likely to be accepted if it is easy to read and understand. Moreover, there are many shortcomings in your study. You can find my comments on the file attached. Your are therefore requested to revise your manuscript and fix the grammatical errors and improve the overall readability of the text.

---

## [Author Response · Author response to Decision Letter 1]

6 Feb 2021

A rebuttal letter that responds to each point raised by the academic editor and reviewer(s). You should upload this letter as 1.A separate file labeled 'Response to Reviewers'.

2.A marked-up copy of your manuscript that highlights changes made to the original version. You should upload this as a separate file labeled 'Revised Manuscript with Track Changes'.

---

## [Editor Report · Decision Letter 2]

23 Mar 2021

PONE-D-20-31879R2

ETHNO-VETERINARY PRACTICES FOR THE TREATMENT OF ANIMAL DISEASES IN NEELUM VALLEY, KASHMIR HIMALAYA, PAKISTAN

PLOS ONE

Dear Dr. Rafiq,

Thank you for submitting your manuscript to PLOS ONE. After careful consideration, we feel that it has merit but does not fully meet PLOS ONE’s publication criteria as it currently stands. Therefore, we invite you to submit a revised version of the manuscript that addresses the points raised during the review process.

ACADEMIC EDITOR:

Address all the comments carefully in attached file (Manuscript_AE)

We look forward to receiving your revised manuscript.

Kind regards,

Khawaja Shafique Ahmad, Ph.D.

Academic Editor

PLOS ONE

Journal Requirements:

Additional Editor Comments (if provided):

Carefully revise and address all the comments in attached file.

---

## [Author Response · Author response to Decision Letter 2]

24 Mar 2021

All the suggested changes and improvements have been made accordingly. Response to the reviewer has been developed and uploaded separately as well.

---

## [Editor Report · Decision Letter 3]

31 Mar 2021

ETHNO-VETERINARY PRACTICE FOR THE TREATMENT OF ANIMAL DISEASES IN NEELUM VALLEY, KASHMIR HIMALAYA, PAKISTAN

PONE-D-20-31879R3

Dear Dr. Rafiq,

We’re pleased to inform you that your manuscript has been judged scientifically suitable for publication and will be formally accepted for publication once it meets all outstanding technical requirements.

Kind regards,

Khawaja Shafique Ahmad, Ph.D.

Academic Editor

PLOS ONE

Additional Editor Comments (optional):

Check minor grammatical mistakes throughout the manuscript. Also cross-check scientific names.
---

## [Editor Report · Acceptance letter]

5 Apr 2021

PONE-D-20-31879R3 

ETHNO-VETERINARY PRACTICE FOR THE TREATMENT OF ANIMAL DISEASES IN NEELUM VALLEY, KASHMIR HIMALAYA, PAKISTAN 

Dear Dr. Rafique Khan:

I'm pleased to inform you that your manuscript has been deemed suitable for publication in PLOS ONE. Congratulations! Your manuscript is now with our production department. 

Kind regards, 

on behalf of

Dr. Khawaja Shafique Ahmad 

Academic Editor

PLOS ONE